# PPI-hotspot[ID] for detecting protein–protein interaction hot spots from the free protein structure

Yao Chi Chen*, Karen Sargsyan*, Jon D Wright[†], Yu-Hsien Chen, Yi-Shuian Huang, Carmay Lim*[†]

Institute of Biomedical Sciences, Academia Sinica, Taipei, Taiwan

**Abstract** Experimental detection of residues critical for protein–protein interactions (PPI) is a time-consuming, costly, and labor-intensive process. Hence, high-throughput PPI-hot spot prediction methods have been developed, but they have been validated using relatively small datasets, which may compromise their predictive reliability. Here, we introduce PPI-hotspot[ID], a novel method for identifying PPI-hot spots using the free protein structure, and validated it on the largest collection of experimentally confirmed PPI-hot spots to date. We explored the possibility of detecting PPI-hot spots using (i) FTMap in the PPI mode, which identifies hot spots on protein–protein interfaces from the *free* protein structure, and (ii) the interface residues predicted by AlphaFold-Multimer. PPI-hotspot[ID] yielded better performance than FTMap and SPOTONE, a webserver for predicting PPI-hot spots given the protein sequence. When combined with the AlphaFold-Multimer-predicted interface residues, PPI-hotspot[ID] yielded better performance than either method alone. Furthermore, we experimentally verified several PPI-hotspot[ID]-predicted PPI-hot spots of eukaryotic elongation factor 2. Notably, PPI-hotspot[ID] can reveal PPI-hot spots not obvious from complex structures, including those in *indirect* contact with binding partners. PPI-hotspot[ID] serves as a valuable tool for understanding PPI mechanisms and aiding drug design. It is available as a web server (https://ppihotspotid.limlab.dnsalias.org/) and open-source code (https://github.com/wrigjz/ppihotspotid/).

**\*For correspondence:**
backy2010.chen@gmail.com
(YCC);
karen.sarkisyan@gmail.com (KS);
carmay@gate.sinica.edu.tw (CL)

**Present address:** [†]Immunwork, Inc, Taipei, Taiwan

## eLife assessment

The article presents a machine-learning method to predict protein hotspot residues. The validation is **incomplete**, along with the misinterpretation of the results with other current methods like FTMap.

## Introduction

Protein–protein interactions (PPIs) play a crucial role in cellular physiology, and their dysregulation is associated with various diseases (*David et al., 2012*) such as cancer (*Nero et al., 2014*), infectious diseases, and neurodegenerative diseases (*Blazer and Neubig, 2009*). Identifying residues critical for PPIs (termed PPI-hot spots) is important for elucidating protein function and designing targeted biomedical interventions (*Cukuroglu et al., 2014*; *Rosell and Fernández-Recio, 2018*). Conventionally, PPI-hot spots are defined as residues whose mutations to alanine cause ≥2 kcal/mol drop in the protein binding free energy (*Clackson and Wells, 1995*; *Bogan and Thorn, 1998*; *DeLano, 2002*; *Li et al., 2004*; *Keskin et al., 2005*; *Moreira et al., 2007*). However, this definition, based on measuring the binding free energy change upon mutation to alanine, limits the number of experimentally determined PPI-hot spots. Hence, PPI-hot spots have been more broadly defined to include residues whose mutations, not necessarily to alanine, significantly impair/disrupt PPIs (*Fischer et al., 2003*; *Chen et al., 2022*), as detected by experimental methods such as coimmunoprecipitation and

yeast two-hybrid screening. As each mutant must be purified and analyzed separately, experimental detection of PPI-hot spots is time-consuming, costly, and labor-intensive.

To enable large-scale detection of PPI-hot spots, high-throughput PPI-hot spot prediction methods have been developed. They generally fall into two categories (*Rosário-Ferreira et al., 2022*): (1) methods that compute the binding energy/free energy difference between the wild-type protein and a mutant using classical force fields or empirical scoring functions (*Moreira et al., 2007*; *Massova and Kollman, 1999*; *Huo et al., 2002*; *Guerois et al., 2002*; *Kortemme and Baker, 2002*; *González-Ruiz and Gohlke, 2006*; *Grosdidier and Fernández-Recio, 2008*; *Yogurtcu et al., 2008*; *Barlow et al., 2018*; *Ibarra et al., 2019*). (2) Methods that employ classifiers such as nearest neighbor, support vector machines, decision trees, Bayesian/neural networks, random forest, and ensemble machine-learning models using various features including conservation, secondary structure, solvent-accessible surface area (SASA), and atom density (*Rosário-Ferreira et al., 2022*; *Darnell et al., 2007*; *Cho et al., 2009*; *Assi et al., 2010*; *Xia et al., 2010*; *Lise et al., 2011*; *Wang et al., 2012*; *Ye et al., 2014*; *Munteanu et al., 2015*; *Melo et al., 2016*; *Moreira et al., 2017*; *Qiao et al., 2018*; *Sitani et al., 2021*; *Ovek et al., 2022*). Most of the PPI-hot spot prediction methods rely on the protein complex structure and some are accessible via webservers; for example, Hotpoint (*Tuncbag et al., 2010*), KFC2 (*Zhu and Mitchell, 2011*), PredHS (*Deng et al., 2014*), and PredHS2 (*Wang et al., 2018a*). Fewer methods use only the *free* protein structure (*Higa and Tozzi, 2009*; *Zerbe et al., 2012*; *Ozbek et al., 2013*; *Agrawal et al., 2014*; *Kozakov et al., 2015*) or sequence (*Qiao et al., 2018*; *Ofran and Rost, 2007*; *Chen et al., 2013*; *Nguyen et al., 2013*; *Huang and Zhang, 2016*; *Hu et al., 2017*; *Jiang et al., 2017*; *Liu et al., 2018*; *Preto and Moreira, 2020*; *Yao et al., 2022*), and SPOTONE (hot SPOTs ON protein complexes with Extremely randomized trees) is available as a web server. SPOTONE (*Preto and Moreira, 2020*) predicts PPI-hot spots from the protein sequence using residue-specific features such as atom type, amino acid (aa) properties, secondary structure propensity, and mass-associated values to train an ensemble of extremely randomized trees.

The PPI-hot spot prediction methods have mostly been trained, validated, and tested on data from the Alanine Scanning Energetics database (ASEdb) (*Thorn and Bogan, 2001*) and/or the Structural Kinetic and Energetic database of Mutant Protein Interactions (SKEMPI) 2.0 database (*Jankauskaite et al., 2019*). However, antibody–antigen interactions have different sequence and structural characteristics compared to non-antibody PPIs (*Wang et al., 2018b*). Therefore, our focus is exclusively on *non*-antibody proteins in this study. The ASEdb contains 96 PPI-hot spots from 26 proteins. SKEMPI 2.0, which includes single-point mutations not necessarily to alanine that decrease the protein binding free energy by ≥2 kcal/mol, has 343 PPI-hot spots from 117 distinct proteins, 40 of which overlap with ASEdb. Altogether, ASEdb and SKEMPI contain 399 distinct PPI-hot spots in 132 proteins. To increase this number of distinct PPI-hot spots, we have expanded the definition of PPI-hot spots to include mutations in UniProtKB (*UniProt Consortium, 2018*) that have been manually curated as significantly impairing/disrupting PPIs (*Chen et al., 2022*). This expanded definition led to the creation of the PPI-Hotspot[DB] database, which contains 4039 experimentally determined PPI-hot spots in 1893 proteins. To calibrate PPI-hot spot prediction methods using the free protein structure, a benchmark was derived from PPI-Hotspot[DB] (*Chen et al., 2022*). This benchmark called PPI-Hotspot+PDB[BM] contains nonredundant proteins with free structures and known PPI-hot spots. The proteins in PPI-Hotspot+PDB[BM] share <60% sequence identity, which has been shown to be a reasonable threshold for grouping domains with similar functions (*Chen et al., 2022*).

Our aim is to develop a method for identifying PPI-hot spots in non-antibody proteins using free protein structures. First, we updated the PPI-Hotspot+PDB[BM] benchmark and constructed a dataset comprising 158 nonredundant proteins with free structures harboring 414 experimentally known PPI-hot spots and 504 PPI-nonhot spots (see 'Materials and methods'). Using this dataset, we applied an automatic machine-learning framework that automates the machine-learning pipeline to detect PPI-hot spots using the aa type as well as structural, energetic, and evolutionary features of each residue in a protein. The resulting prediction model, named PPI-hotspot[ID], identifies PPI-hot spots using an ensemble of classifiers and only four residue features (conservation, aa type, SASA, and gas-phase energy, $\Delta G^{gas}$). We explored the possibility of detecting PPI-hot spots using the FTMap server in the PPI mode, which identifies hot spots on protein–protein interfaces from *free* protein structures (*Kozakov et al., 2015*). These hot spots are identified by consensus sites – regions that bind multiple probe clusters (*Zerbe et al., 2012*; *Kozakov et al., 2015*; *Kozakov et al., 2011*). Such

regions are deemed to be important for any interaction involving that region of the target, independent of partner protein (*Zerbe et al., 2012*). PPI-hot spots were identified as residues in van der Waals (vdW) contact with probe ligands within the largest consensus site containing the most probe clusters. We also explored the possibility of detecting PPI-hot spots using the interface residues predicted by AlphaFold-Multimer (*Evans et al., 2021*), which has been shown to outperform current docking methods in predicting protein–protein complexes. Finally, we illustrated the utility of PPI-hotspot[ID] by applying it to detect PPI-hot spots of eukaryotic elongation factor 2 (eEF2), a translation factor essential for peptide elongation, and experimentally verified the predictions.

## Results

### Evaluating the performance of PPI-hot spot detection methods

The goal of PPI-hotspot[ID] is to detect true PPI-hot spots rather than true PPI-nonhot spots in proteins. Hence, we assessed the performance of PPI-hotspot[ID] by computing the sensitivity/recall (the fraction of true PPI-hot spots correctly identified),

$$\text{Sensitivity} = \frac{TP}{TP + FN} \tag{1}$$

the fraction of predicted PPI-hot spots that are true PPI-hot spots; that is,

$$\text{Precision} = \frac{TP}{TP + FP} \tag{2}$$

and the F1-score, which combines recall and precision:

$$\text{F}_1 = \frac{2 \times \left(sensitivity \times precision\right)}{\left(sensitivity + precision\right)} = \frac{2TP}{2TP + FP + FN} \tag{3}$$

Since our dataset also contains true PPI-nonhot spots, we calculated the specificity (the fraction of true PPI-nonhot spots correctly identified):

$$\text{Specificity} = \frac{TN}{TN + FP} \tag{4}$$

In *Equations 1–4*, TP (true positives) or TN (true negatives) is the number of correctly predicted PPI-hot spots or PPI-nonhot spots, and FP (false positives) or FN (false negatives) is the number of wrongly predicted PPI-hot spots or PPI-nonhot spots.

### Performance of PPI-hotspot[ID] vs. FTMap and SPOTONE

We compared the performance of PPI-hotspot[ID], FTMap (*Kozakov et al., 2015*), and SPOTONE (*Preto and Moreira, 2020*) using a dataset containing 414 true PPI-hot spots and 504 nonhot spots. *Source data 1* lists the UniProt codes of the PPI-hot spot-containing protein and its binding partner, the PDB code(s) and chain of the free protein structure, the UniProt and PDB numbering of the PPI-hot spot, the wild-type→mutant residue, the corresponding protein binding free energy change and source given by the PubMed reference number, and the PPI-hotspot[ID] assignment, where P indicates PPI-hot spot and N indicates nonhot spot. *Source data 2* lists the UniProt codes of the PPI-hot spot-containing protein A and binding partner protein B, the PDB code-chain and length of the free and bound protein A structures, the PDB code-chain of the bound protein B structure, the sequence identity between free and bound protein A structures, and the PPI-hot spots of protein A. Note that the 414 true PPI-hot spots represent only 2% of the total number of residues (21,722) in the 158 proteins.

Given the free protein structure, PPI-hotspot[ID] and SPOTONE (*Preto and Moreira, 2020*) predict PPI-hot spots based on a probability threshold (>0.5). FTMap, in the PPI mode, detects PPI-hot spots as consensus sites/regions on the protein surface that bind multiple probe clusters (*Kozakov et al., 2011*). Residues in vdW contact with probe molecules within the largest consensus site were compared with PPI-hotspot[ID]/SPOTONE predictions. Residues *not* classified as PPI-hot spots by each method were considered as PPI-nonhot spots. *Table 1* summarizes the results for our dataset, with the F1 score in parentheses representing the mean validation F1 score. Compared to FTMap/SPOTONE,

**Table 1.** Performance of the PPI-hotspot[ID] vs. FTMap and SPOTONE.

| Method | PPI-hotspot[ID] | FTMap | SPOTONE |
|---|---|---|---|
| TP | 278 | 30 | 40 |
| FN | 136 | 384 | 374 |
| TN | 417 | 487 | 481 |
| FP | 87 | 17 | 23 |
| Sensitivity | 0.67 | 0.07 | 0.10 |
| Precision | 0.76 | 0.64 | 0.64 |
| F1 | 0.71 (0.66)* | 0.13 | 0.17 |
| Specificity | 0.83 | 0.97 | 0.95 |

Each method was tested using the same dataset comprising 414 experimentally known PPI-hot spots (TP + FN) and 504 PPI-nonhot spots (TN + FP).
TP = true positive; FP = false positive; TN = true negative; FN = false negative.
*The F1 score in parentheses corresponds to the validation F1 score.

PPI-hotspot[ID] detected a much higher fraction of true positives (0.67 vs. 0.07/0.10) and achieved a significantly higher F1 score (0.71 vs. 0.13/0.17).

To elucidate the differences between the PPI-hot spots predicted by PPI-hotspot[ID] and those by FTMap or SPOTONE, we compared their respective true-positive predictions. The Venn diagram in *Table 1* shows a substantial overlap in true positives between FTMap or SPOTONE and PPI-hotspot[ID]: FTMap shared 23/30 true positives with PPI-hotspot[ID], whereas SPOTONE shared 34/40 with PPI-hotspot[ID], but only 3 with FTMap. Only three true positives were predicted by all three methods. PPI-hotspot[ID] identified many true positives that were not detected by FTMap or SPOTONE probably because it employed aspects not considered by FTMap or SPOTONE such as the gas-phase energy, $\Delta G^{gas}$ (see 'Discussion'). Furthermore, SPOTONE defined true negatives as residues whose mutation to alanine led to protein binding free energy changes ($\Delta\Delta G^{bind}$) of ≤2.0 kcal/mol, whereas we defined true negatives as residues whose alanine/nonalanine mutations resulted in negligible protein binding free energy changes ($\Delta\Delta G^{bind} < 0.5$ kcal/mol) or did not perturb PPIs in immunoprecipitation or GST pull-down assays (see 'Materials and methods').

## Interface vs. noninterface PPI-hot spots

We can estimate the fraction of PPI-hot spots at the protein interface for 74 of the 158 nonredundant proteins in our dataset with complex structures. These 74 proteins harboring 243 true PPI-hot spots form 78 PPI pairs. Using the UniProt codes of each protein and its binding partner, we identified all complex structures in the PDB. Based on the complex structures of each PPI pair, we classified a PPI-hot spot as an interface one if it formed hydrogen bonds or vdW contacts with the partner protein (*Laskowski et al., 2018*); otherwise, it was deemed a *non*interface PPI-hot spot. Among the 243 true PPI-hot spots, 67 (27.6%) lacked such contacts across the protein interface. For these 74 proteins, PPI-hotspot[ID] predicted 240 PPI-hot spots, out of which 43 (18%) are *non*interface PPI-hot spots, SPOTONE identified only five noninterface PPI-hot spots, whereas FTMap did not predict any. For example, the complex structure of interleukin-4 bound to interleukin-4 receptor subunitα (PDB: 1IAR) (*Hage et al., 1999*) in *Figure 1* revealed three interface PPI-hot spots (E13, R92, and N93) and two noninterface ones (K127 and Y128). Based on the free structure of interleukin-4 (PDB: 1BBN) (*Powers et al., 1992*), PPI-hotspot[ID] identified all five true positives, SPOTONE detected an interfacial PPI-hot spot (N93), whereas FTMap failed to identify any true positives.

## Performance of AlphaFold-Multimer, PPI-hotspot[ID] and their combination in predicting PPI-hot spots

To assess the possibility of detecting PPI-hot spots using the interface residues predicted by AlphaFold-Multimer (*Evans et al., 2021*) as PPI-hot spots when complex structures are unavailable, we focused on 48 'unsolved' AB complex structures involving 47 proteins in the PPI-Hotspot+PDB[BM(1.1)], as one of the

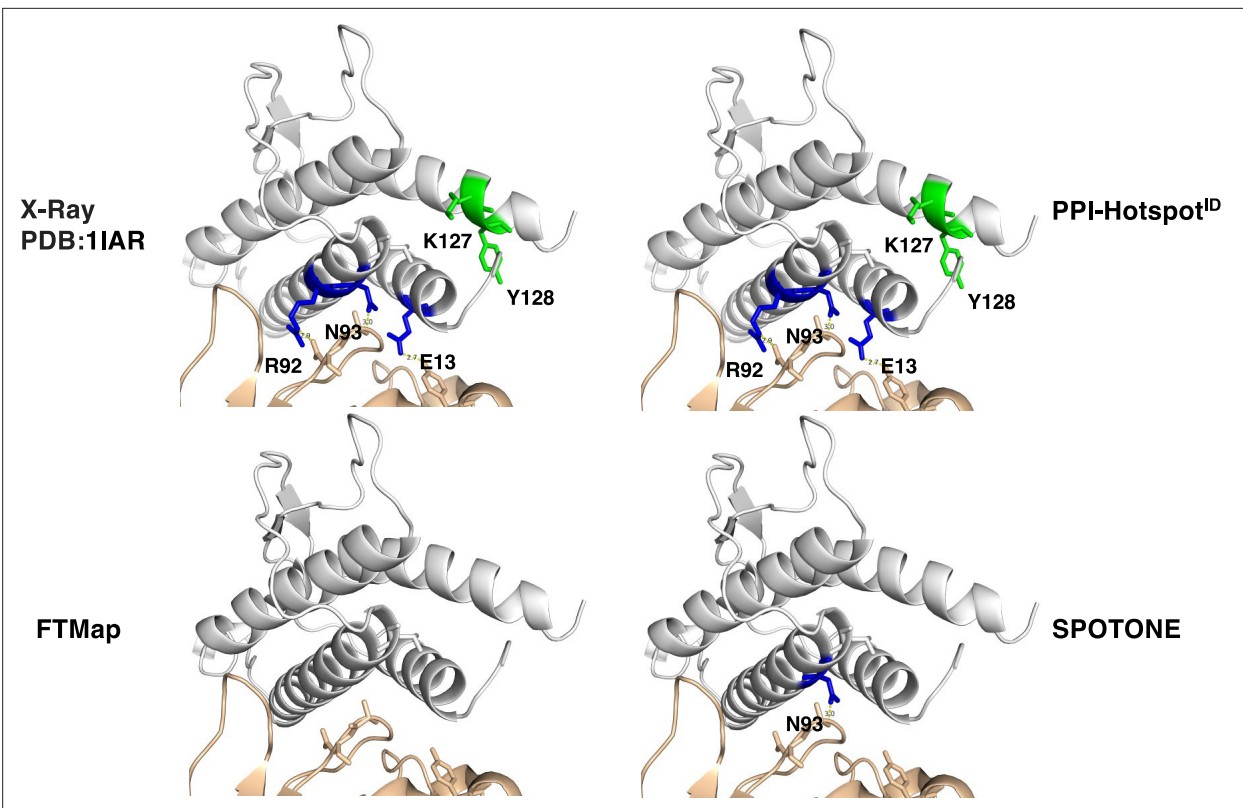

**Figure 1.** Interface vs. noninterface PPI-hot spots. (Top left) The X-ray structure (PDB: 1IAR) of interleukin-4 (gray) in complex with interleukin-4 receptor subunit alpha (wheat) with five PPI-hot spots; interface PPI-hot spots (E13, R92, and N93) are in blue and the noninterface ones (K127 and Y128) are in green. The PPI-hot spot numberings are based on the interleukin-4 free structure (PDB: 1BBN). The correct predictions of PPI-hotspot[ID] (top right), FTMap (bottom left), and SPOTONE (bottom right) are mapped to the corresponding residues of the complex structure.

proteins, human neurotrophin (UniProtID P20783, PDB 1nt30A) interacted with two different proteins (UniProtID Q16288 and P17643). These 48 *unsolved* complex structures contain 90 PPI-hot spots and 45 nonhot spots. We employed the protein A structure sequence from the PPI-Hotspot+PDB[BM(1.1)] and the entire protein B sequence from UniProtKB (*UniProt Consortium, 2018*) as inputs for the AlphaFold-Multimer module in ColabFold (*Mirdita et al., 2022*). This generated model structures for each AB complex. Interface residues were defined based on the AMBER-relaxed model structure with the highest pTM score using a cutoff distance of 5 Å reflecting residues in close contact. Interface residues were predicted as PPI-hot spots and noninterface residues as nonhot spots.

In identifying PPI-hot spots using PPI-hotspot[ID], we first excluded 90 true PPI-hot spots and 45 nonhot spots belonging to 47 proteins lacking complex PDB structures from our dataset. We then used an automatic machine-learning framework to train an ensemble of machine-learning models using four features ($k^C$, aa residue type, $SASA_i$, and $\Delta G_i^{gas}$) on the true PPI-hot spots and nonhot spots in the remaining 111 proteins in our dataset. The final ensemble model was used to identify PPI-hot spots in the 47 proteins lacking complex structures in our dataset. The resulting sensitivity (0.58) and F1 score (0.66) in *Table 2* were lower than those in *Table 1* using the full dataset. Nevertheless, they were greater than those achieved using AlphaFold-Multimer-predicted interface residues as PPI-hot spots (0.41 and 0.54). When we combined the PPI-hotspot[ID]-predicted PPI-hot spots with the AlphaFold-Multimer-predicted interface residues, the resulting sensitivity (0.70) and F1 values (0.72) were higher than those obtained by each method alone. This indicates that PPI-hotspot[ID] can identify true PPI-hot spots that reside outside the protein–protein interface.

## Experimental verification of PPI-hotspot[ID]'s predictions in human eEF2

We experimentally verified predictions made by PPI-hotspot[ID] by using it to detect the PPI-hot spots of eEF2, an essential translation factor that hydrolyzes GTP to catalyze peptide elongation. Binding of

**Table 2.** Performance of AlphaFold-Multimer, PPI-hotspot[ID], and their combination for 48 'unsolved' complex structures.

| Method | AlphaFold2-Multimer | PPI-hotspot[ID] | AlphaFold2-Multimer+PPI-hotspot[ID] |
|---|---|---|---|
| TP | 37 | 52 | 63 |
| FN | 53 | 38 | 27 |
| TN | 35 | 29 | 24 |
| FP | 10 | 16 | 21 |
| Sensitivity | 0.41 | 0.58 | 0.70 |
| Precision | 0.79 | 0.77 | 0.75 |
| F1 | 0.54 | 0.66[*] | 0.72 |
| Specificity | 0.78 | 0.64 | 0.53 |

Each method was tested using the same dataset comprising 90 experimentally known PPI-hot spots (TP+FN) and 45 PPI-nonhot spots (TN+FP) in 48 protein complexes with no known structures.

TP = true positive; FP = false positive; TN = true negative; FN = false negative.

[*]No validation F1 score is provided since AutoGluon was used to train an ensemble of machine-learning models on a dataset that excludes the 48 'unsolved' complex structures (see text).

cytoplasmic polyadenylation element-binding protein-2 (CPEB2) to eEF2 may interfere with conformational changes of eEF2 on ribosomes, thereby affecting the efficiency of eEF2-mediated GTP hydrolysis, and slowing down translation of hypoxia-inducible factor (HIF)–1α mRNA (*Chen and Huang, 2012*). No eEF2-CPEB2 complex structure has been solved, but a 5 Å electron microscopy structure of eEF2 (PDB 4v6x-A) (*Anger et al., 2013*) is available. Using the CPEB2 N-terminus for a yeast two-hybrid screen, a positive clone containing the eEF2 residues 717–803 had been identified and subsequent co-IP assay revealed a CPEB2-binding domain comprising eEF2 residues 743–817 (*Chen and Huang, 2012*). Thus, we focused on this domain, which shares ≤20% sequence identity with the 158 nonredundant proteins in our dataset, in predicting PPI-hot spots. Based on the free eEF2 structure (PDB 4v6x-A) (*Anger et al., 2013*), PPI-hotspot[ID] predicted F794 as the PPI-hot spot with the highest probability of 0.67. So, we chose to test F794 and seven other predicted PPI-hot spots (L763, R767, G768, G778, T779, R801, A808) that were >12 Å from F794, as well as four predicted PPI-*non*hot spots (E773, P789, V790, Q807).

To validate PPI-hotspot[ID]'s predictions, we mutated the aforementioned predicted PPI-hot spots and PPI-*non*hot spots in mouse eEF2 (meEF2), which shares 99% sequence identity with human eEF2. The generated eEF2 mutants (L763A, [766]AAA[768], E773Q, [778]AAA[780], [789]AA[790], F794A, R801A, Q807E, A808S, and D815A), along with wild-type eEF2 and negative control (enhanced green fluorescent protein [EGFP]), were then screened for interaction with CPEB2 by co-immunoprecipitation (co-IP). This assay identified F794 as a critical eEF2 residue for binding to CPEB2. To confirm the initial screening result, we selected three mutants ([778]AAA[780], F794A, and D815A designated as mut1, mut2, and mut3) for further analysis (*Figure 2a*). The interaction of wild-type and mutant eEF2 with CPEB2 was analyzed again by reciprocal co-IP. The results in *Figure 2b* show that the F794A mutation (mut2) abolished binding to CPEB2.

Next, we investigated whether disrupting the association between CPEB2 and eEF2 affects HIF-1α expression in vivo. Because eEF2 is an essential and abundant translational factor, its ectopic expression alone was insufficient to override the function of endogenous eEF2. Thus, we tested the F794A mutant under knockdown of endogenous eEF2 condition. HeLa cells were transfected with plasmids lacking (siCtrl) or containing a short hairpin sequence for human eEF2 (siheEF2) and subjected to puromycin selection. Both shRNA sequences, specifically knocking down human but not mouse eEF2, decreased endogenous eEF2 after 4 days (*Figure 2c*). HeLa cells were then transfected with the siheEF2 and flag-meEF2 (wild-type, mut2, or mut3) plasmids and subjected to puromycin and G418 selection for 4 days. Cells that survived were incubated with S[35]-methionine/cysteine to metabolically label synthesized proteins. The expression of the F794A or D815A mutant did not affect general protein synthesis (*Figure 2d*). However, the level of HIF-1α, but not CPEB2 or β-actin, was selectively increased in HeLa cells reconstituted with the F794A mutant (*Figure 2e*). *Figure 2—figure*



**Figure 2.** Evaluation of the predicted CPEB2-interacting amino acid residues in eEF2. (**a**) Salient features of mouse eEF2, showing the various domains and the mutated amino acids in domain V. mut 1, G778A, T779A, and P780A; mut 2, F794A; mut 3, D815A. (**b**) Reciprocal co-immunoprecipitation (co-IP). The 293T cells expressing myc-CPEB2 along with wt or mutant flag-eEF2 or control GFP were harvested and then precipitated with flag or myc IgG. The precipitated substances were used for western blotting with myc and flag antibodies. IP, immunoprecipitation; IB, immunoblotting; IgG H.C., IgG heavy chain. (**c**) HeLa cells transfected with the plasmid expressing shRNA against human eEF2 (siheEF2) were harvested after 4-day puromycin selection for western blotting. HeLa cells transfected with the eEF2 knockdown plasmid along with flag-tagged wt or mutant mouse eEF2 after 4-day selection with puromycin and G418 were used for (**d**) $S^{35}$-met/cys-labeling of synthesized proteins or (**e**) western blotting with the denoted antibodies. The normalized HIF-1α protein level (HIF-1α/β-actin signal) was calculated and expressed as mean ± SEM from three independent experiments. Two-tailed Student's t-test, *<0.05.

The online version of this article includes the following source data and figure supplement(s) for figure 2:

**Source data 1.** Containing uncropped images of the membranes for *Figure 2b, c and e* and phosphoimager file for *Figure 2d*.

**Source data 2.** Containing the original files of the full raw unedited blots.

*Figure 2 continued on next page*

*Figure 2 continued*

**Source data 3.** Listing primer sequences.

**Figure supplement 1.** Uncropped immunoblot images.

*supplement 1* shows full-length gels and blots in *Figure 2*, and *Figure 2—source data 1* shows the uncropped immunoblot images, and *Figure 2—source data 2* contains raw unedited blots. Thus, the eEF2 F794A mutation influences the translation of CPEB2-targeted HIF-1α mRNA without affecting general translation function.

## Discussion

Identifying PPI-hot spots is challenging especially when the complex structure is lacking. A key hurdle is the lack of experimental data on PPI-hot spots, which hampers the training of accurate machine-learning models for their prediction. Here, we introduced two novel elements that have helped to identify PPI-hot spots using the unbound structure. First, we have constructed a dataset comprising 414 experimentally known PPI-hot spots and 504 nonhot spots, and *carefully* checked that PPI-hot spots have no mutations resulting in $\Delta\Delta G^{bind} < 0.5$ kcal/mol, whereas nonhot spots have no mutations resulting in $\Delta\Delta G^{bind} \geq 0.5$ kcal/mol or impact binding in immunoprecipitation or GST pull-down assays (see 'Materials and methods'). In contrast, SPOTONE (*Preto and Moreira, 2020*) employed nonhot spots defined as residues that upon alanine mutation resulted in $\Delta\Delta G^{bind} < 2.0$ kcal/mol. Notably, previous PPI-hot spot prediction methods did not employ PPI-hot spots whose mutations have been curated to significantly impair/disrupt PPIs in UniProtKB (see 'Introduction'). Second, we introduced novel features derived from *unbound* protein structures such as the gas-phase energy of the target protein relative to its unfolded state. The importance test results indicated the gas-phase energy as an important feature. This finding can be rationalized by considering how PPI-hot spots make significant contributions to the overall binding free energy, $\Delta G^{bind}$. PPI-hot spots can enhance favorable enthalpic contributions to the $\Delta G^{bind}$ through hydrogen bonds or vdW contacts across the protein's interface. This makes them energetically *unstable* in the absence of the protein's binding partner and solvent; hence, the gas-phase energy was found to be an important input feature. Alternatively, PPI-hot spots can counteract unfavorable entropic loss upon protein binding by maintaining an optimal binding scaffold; hence, they are energetically *stable*.

Methods that rely on complex structures generally predict residues that make multiple contacts across the protein–protein interface as PPI-hot spots. Some of these methods assume that PPI-hot spots are exclusively located at the interface and aim to spot them among the interface residues (*Wang et al., 2018a*). In contrast, PPI-hotspot[ID] leverages evolutionary conservation, residue type, and stability principles based on the free protein structure to detect PPI-hot spots, including those lacking direct contact with the partner protein. Such noninterface PPI-hot spots may serve to maintain an optimal scaffold for protein binding and are not uncommon: from our analysis of the 243 true PPI-hot spots in proteins with the complex structures, we found 67 'noninterface' PPI-hot spots with no hydrogen bonds and/or vdW contacts across the protein

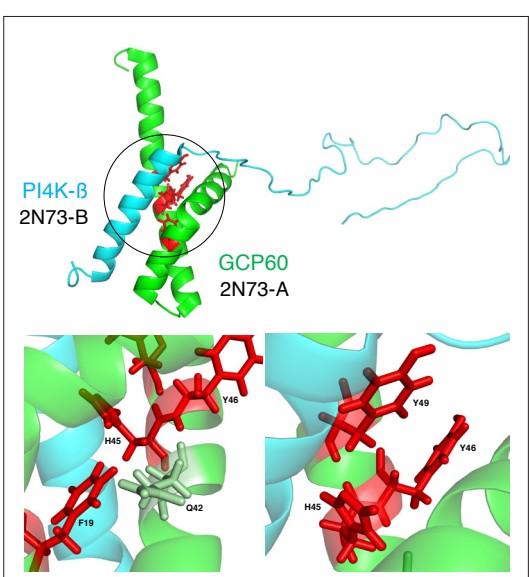

**Figure 3.** Interface and noninterface PPI-hot spots of Golgi resident protein (GCP60). (Top) The structure (PDB 2N73) (*Wright et al., 2024*) of GCP60 (green) in complex with PI4K-β (cyan) with the GCP60–PI4K-β interface encircled. (Bottom) The four experimentally known PPI-hot spots of GCP60 are shown in red. H45 and Y49 form hydrogen bonds across the interface with PI4K-β. Although F19 and Y46 do not directly contact PI4K-β, F19 is in van der Waals (vdW) contact with Q42, which in turn forms vdW contacts with H45, whereas Y46 is in vdW contact with both H45 and Y49.

interface. PPI-hotspot[ID] identified 43 of these 67 noninterface PPI-hot spots. An illustrative example is the binding of Golgi resident protein (GCP60) with phosphatidylinositol 4-kinase β (PI4K-β). PPI-hotspot[ID] correctly predicted all four experimentally known GCP60 PPI-hot spots including F19 and Y46, which do not form hydrogen bonds across the interface with PI4K-β (*Figure 3*). These results highlight the ability of PPI-hotspot[ID] to identify PPI-hot spots involved in indirect interactions with partner proteins.

Proteins typically interact with multiple partners, but their PPI-hot spots may have been experimentally characterized for only a few partners. In some cases where PPI-hotspot[ID] predicted residues that were absent in the PPI-Hotspot+PDB[BM(1.1)] as PPI-hot spots, the protein's complex structures with other binding partners show intermolecular hydrogen bonds between PPI-hotspot[ID]-predicted residues and residues of the respective partner proteins. This suggests that some of the PPI-hotspot[ID]-predicted residues might be potential PPI-hot spots for other binding partners. For example, the death domain of CRADD (caspase-recruitment domain and death domain-containing adaptor protein) contains 7 experimentally known PPI-hot spots (N121, Q125, Y146, R147, K149, V156, Q169) critical for its interaction with PIDD (p53-induced death domain-containing protein). Based on the free crystal structure of CRADD (PDB 2O71-A) (*Park and Wu, 2006*), PPI-hotspot[ID] correctly predicted three true positives (Y146, R147, and Q169), as well as G128. In the oligomeric structure (PDB 2OF5) (*Park et al., 2007*) of seven CRADD proteins in complex with five PIDD proteins, G128 shows no hydrogen-bonding interactions, but its neighbor, L127, forms a backbone – side chain hydrogen bond with R147 in another CRADD chain (*Figure 4*). A positively charged G128R mutation would repel the nearby positively charged R147 in another CRADD chain, thus disrupting the CRADD–CRADD interface and decreasing CRADD's affinity for PIDD. Experimental data showed that the G128R CRADD mutant did not co-immunoprecipitate the PIDD death domain, and patients who have non-syndromic mental retardation possess the G128R mutant (*Puffenberger et al., 2012*). Thus, PPI-hotspot[ID] could unveil a PPI-hot spot, G128, that is not apparent from the 2OF5 complex structure: although G128 does not directly interact with PIDD, its mutation, especially to an Arg, might perturb the CRADD–CRADD interface and thus CRADD's oligomeric structure and binding affinity for PIDD.

The ability of PPI-hotspot[ID] to detect PPI-hot spots provides biologists with a useful tool, as alanine-scanning mutagenesis and protein–protein complex structure determination to identify PPI-hot spots

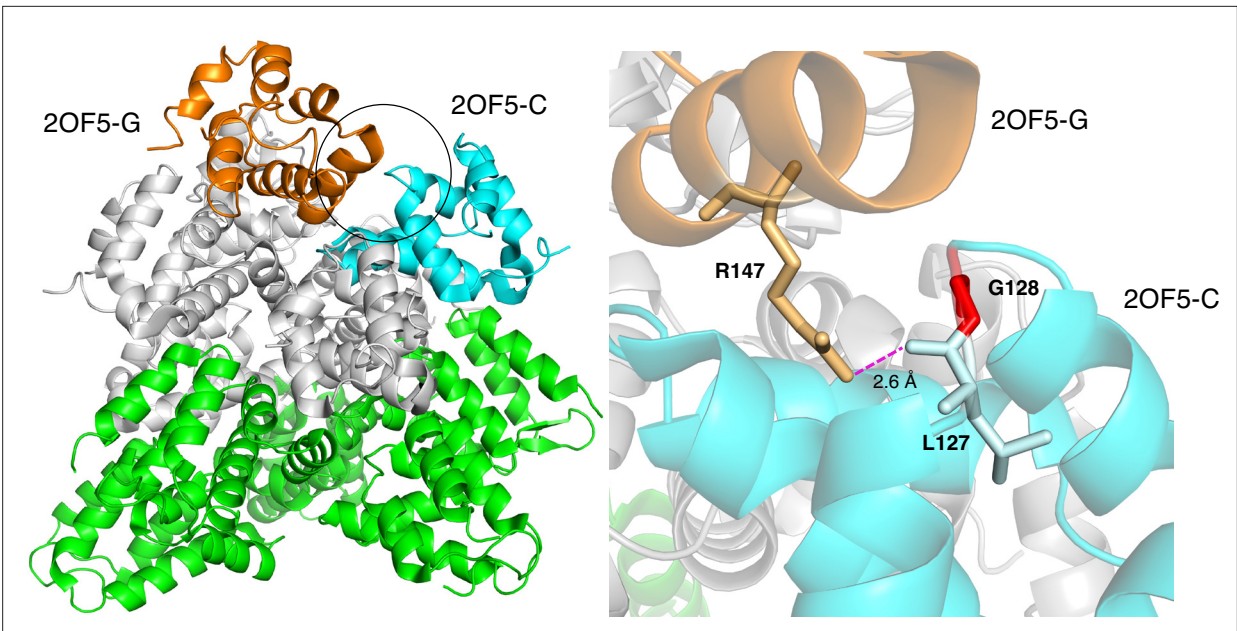

**Figure 4.** Based on the free CRADD X-ray structure (PDB 2O71-A), PPI-hotspot[ID] predicted G128 as a PPI-hot spot for CRADD–CRADD interactions. (Left) The structure (PDB 2OF5) (*Park et al., 2007*) of seven CRADD proteins in complex with fuve PIDD proteins. The circle shows the CRADD–CRADD interface between chains C (cyan) and G (orange), whereas the other five CRADD chains are in gray, and the five PIDD proteins are in green. (Right) G128 (red) in CRADD (chain C) participates indirectly in CRADD–CRADD interactions via a backbone – side chain hydrogen bond between its neighbor, L127, and R147 in another CRADD (chain G).

are laborious, time-consuming, and costly. Conventional methods based on complex structures might miss nonobvious PPI-hot spots with no direct interactions with the protein's partner. AlphaFold-Multimer and future improved protein–protein complex prediction methods require knowledge of interacting partners and independent calculations for each known partner, which reduces the overall efficiency. Moreover, solved/modeled protein–protein complex structures only reveal the interface residues. In contrast, PPI-hotspot[ID] can reveal nonobvious PPI-hot spots as well as potential PPI-hot spots for other protein partners, thus helping to elucidate the different PPI mechanisms.

## Materials and methods

### Dataset: True PPI-hot spots

We updated the PPI-Hotspot+PDB[BM] benchmark by removing two fused protein structures and adding new PPI-hot spots by (i) reviewing references in ASEdb (*Thorn and Bogan, 2001*) to include nonalanine mutations with $\Delta\Delta G^{bind}$ > 2 kcal/mol, and (ii) checking the experimental data of certain mutations in UniProtKB (*UniProt Consortium, 2018*). For example, the PPI-Hotspot+PDB[BM] benchmark included R43A in aprataxin (UniProtID Q7Z2E3), annotated as 'loss of interaction with MDC1', but not K52A, annotated as 'impairs interaction with MDC1'. However, when we checked the experimental data in the UniProtKB reference, the binding bands were absent for both R43A and K52A mutants; therefore, we added K52A as a PPI-hot spot. The updated benchmark, termed PPI-Hotspot+PDB[BM(1.1)], contains 414 PPI-hot spots. Among these, 104 PPI-hot spots in 32 nonredundant proteins are based on mutations resulting in $\Delta\Delta G^{bind} \geq$ 2 kcal/mol from ASEdb (*Thorn and Bogan, 2001*) and SKEMPI2.0 (*Jankauskaite et al., 2019*) with no known mutations resulting in $\Delta\Delta G^{bind}$ < 0.5 kcal/mol. The remaining 310 PPI-hot spots in 128 nonredundant proteins are based on mutations that are manually curated in UniProtKB (*UniProt Consortium, 2018*) to significantly impair/disrupt PPIs. Two of the proteins have PPI-hot spots from ASEdb/SKEMPI2.0 and UniProtKB, resulting in a total of 158 nonredundant proteins with free structures harboring 414 PPI-hot spots.

### True PPI-nonhot spots

To obtain PPI-nonhot spots for the 158 nonredundant proteins with true PPI-hot spots, we identified residues from ASEdb (*Thorn and Bogan, 2001*) and SKEMPI2.0 (*Jankauskaite et al., 2019*) databases where mutations to alanine/nonalanine resulted in protein $\Delta\Delta G^{bind}$ < 0.5 kcal/mol. We also identified residues in the UniProtKB where mutations to alanine/nonalanine were curated *not* to perturb PPIs. We *manually* checked each reference to ensure that mutations of these residues did not lead to $\Delta\Delta G^{bind}$ changes ≥0.5 kcal/mol or impact binding in immunoprecipitation or GST pull-down assays. PPI-nonhot spots in non-native proteins or regions with missing structures were excluded.

### Input features

To distinguish PPI-hot spots from PPI-nonhot spots, we input sequence, structural, and stability features of each residue in the protein for training various machine-learning classifiers. The input features for each residue $i$ of a protein included its aa type, conservation score, secondary structure, SASA, gas-phase energy, and respective components, polar solvation free energy, and nonpolar solvation free energy. The secondary structure, SASA, and energy components of each residue were computed using the DSSP program (*Kabsch and Sander, 1983*), FreeSasa (*Mitternacht, 2016*), and AmberTools version 20 (*Case, 2020*), respectively, using default parameters.

### Per-residue free energy contributions

For a given free protein structure, the Reduce program (*Word et al., 1999*) was used to add hydrogens and assign the protonation states of ionizable residues. Additional missing heavy and hydrogen atoms were added using the AmberTools version 20 (*Case, 2020*) and the Amber FF19SB forcefield (*Tian et al., 2020*). To eliminate any steric clashes, we performed a conjugate gradients minimization with constraints on the heavy atoms using the Generalized Born model for 500 steps. The resulting structure was used to compute the per-residue energy/free energy contributions using the Molecular Mechanics Poisson–Boltzmann Surface Area module in AmberTools (*Case, 2020*). For each residue $i$ in the protein, we computed the (i) molecular mechanics energy $E_i^{gas} = E_i^{MM,int} + E_i^{MM,vdW} + E_i^{MM,ele}$, where $E_i^{MM,int}$ includes contributions from bonded terms, $E_i^{MM,vdW}$ is the vdW interaction energy, and $E_i^{MM,ele}$ is

the electrostatic interaction energy as well as (ii) the polar ($\Delta G_i^{solv,pol}$) and nonpolar ($\Delta G_i^{solv,npl}$) solvation free energies relative to the corresponding values of residue $i$ in an extended reference state where the residues do not interact with one another (*Chen et al., 2007*).

## Per-residue conservation score

To calculate the conservation score, $k_i^C$, of residue $i$ in a protein, we implemented a method similar to ConSurf (*Glaser et al., 2003*; *Landau et al., 2005*) to run in parallel with the energy evaluation code. First, we searched the UNIREF-90 database (*Wu et al., 2006*) using HMMER (*Johnson et al., 2010*) to find sequences similar to the target sequence. Near-duplicates were removed by clustering matched sequences with ≥95% pairwise sequence identity using CD-hit (*Li and Godzik, 2006*) and keeping only one representative. Since HMMER (*Johnson et al., 2010*) may only find good matches for a small proportion of the target sequence, we compared the HMMER sequences with the target sequence. We kept only those with >60% overlap with the target sequence and discarded sequences that were dissimilar (≤35% sequence identity) or nearly identical (≥95% sequence identity). Next, we pairwise aligned the remaining sequences, and if two sequences overlapped by >10% of the sequence, we rejected the shorter sequence. After this filtering process, the resulting HMMER hits were used, or if the number of hits exceeded 300, we selected the top 300 hits. These sequences were then aligned to the target sequence using MAFFT-LINSi (*Nakamura et al., 2018*). We then used the Rate4Site program (*Pupko et al., 2002*) to compute position-specific evolutionary rates from the generated multiple sequence alignment. These rates were normalized and grouped into ConSurf grades ranging from 1 to 9, where $k^C = 1$ represents the most rapidly evolving residues, and $k^C = 9$ indicates the most conserved residues.

## Generating PPI-hot spot predictive model using AutoGluon

We provided all the aforementioned residue features including the conservation score, $k_i^C$, aa type, DSSP secondary structure, $SASA_i$, $E_i^{MM,int}$, $E_i^{MM,vdW}$, $E_i^{MM,ele}$, $E_i^{MM}$, $\Delta G_i^{solv,pol}$, and $\Delta G_i^{solv,npl}$ to the Tabular module in AutoGluon v0.8.2 (https://auto.gluon.ai/stable/index.html). AutoGluon was chosen for model training and validation due to its robustness and user-friendly interface, allowing for the simultaneous and automated exploration of various machine-learning approaches and their combinations. Instead of using a single training set to train the model and a separate test set to evaluate its performance, we employed cross-validation as it utilizes the entire dataset for both training and testing, making efficient use of the limited data on PPI-hot spots and PPI-nonhot spots. AutoGluon-Tabular automatically chose a random partitioning of our dataset into multiple subsets/folds for training and validation. Notably, the training and validation data share insignificant homology as the average pairwise sequence identity in our dataset is 26%. Each fold was used once as a test set, while the remaining folds served as the training set. For each test set, the model's performance was measured using the F1 score.

AutoGluon then trained individual 'base' models, including LightGBM, CatBoost, XGBoost, random forests, extremely randomized trees, neural networks, and K-nearest neighbors. Using the aggregated predictions of the base models as features in addition to the original features, AutoGluon trained multiple 'stacker' models, whose predictions were fed as inputs to additional higher layer stacker models in an iterative process called multilayer stacking. The output layer used ensemble selection to aggregate the predictions of the stacker models. To improve stacking performance, AutoGluon used all the data for both training and validation through repeated $k$-fold bagging of all models at all layers of the stack, where $k$ is determined by best precision. We refer the reader to the original study by *Erickson et al., 2020*, which provides details on the methodology including the types of 'base' models, multilayer stack ensembling, and repeated $k$-fold bagging. Based on the highest mean F1 score, AutoGluon yielded a final PPI-hot spot predictive model that is a weighted regularized ensemble comprising more than a dozen different models (https://auto.gluon.ai/dev/api/autogluon.tabular.models.html).

## Selecting key features

Next, we evaluated the importance of each feature by performing a permutation-based test (part of the AutoGluon package), in which a feature in a column was randomly shuffled across different residues (rows), and the F1 score was evaluated. The importance test results revealed the four most

important residue features, which in order of their importance are (i) $k^C$, (ii) aa residue type, (iii) $SASA_i$, and (iv) $\Delta G_i^{gas}$. These four features were used to train an ensemble of machine-learning models using the entire dataset, consisting of 414 true PPI-hot spots and 504 nonhot spots. The resulting PPI-hot spot prediction model, named PPI-hotspot$^{ID}$, yielded an F1 score comparable to the F1 score obtained using the initial set of 10 features. PPI-hotspot$^{ID}$ was implemented as a freely accessible web server (https://ppihotspotid.limlab.dnsalias.org/; *Wright et al., 2024*) with access to four virtual CPUs and 8 GB of memory. Calculations for a 539-residue protein (PDB 1c2bA) took 35 min. The source code for PPI-hotspot$^{ID}$ is available at https://github.com/wrigjz/ppihotspotid/ (*Wright, 2024*).

## Detecting PPI-hot spots using the AlphaFold-Multimer-predicted interface

In cases where experimental complex structures are unavailable, can the protein–protein complexes modeled by AlphaFold-Multimer (*Evans et al., 2021*) be used to identify PPI-hot spots using the predicted interface residues? To address this, we first identified PPI-hot spots within the PPI-Hotspot+PDB$^{BM(1.1)}$ dataset that lack experimentally determined protein complex structures. Not all the 414 PPI-hot spots in the PPI-Hotspot+PDB$^{BM(1.1)}$ have sequence information and thus UniProtID of the respective binding partners, leaving 360 PPI-hot spots in 135 proteins associated with 155 pairs of PPIs, as some proteins are involved in multiple PPIs. Also, 90 of the 155 PPI pairs have complex structures in the PDB. For the 65 PPI pairs lacking complex structures, 17 pairs contain >1100 residues, exceeding the current size limit of AlphaFold-Multimer (*Evans et al., 2021*). Thus, we generated structural models for the remaining 48 complexes using the AlphaFold-Multimer module in the ColabFold version 1.3.0 (*Mirdita et al., 2022*) with default settings. For each AB complex, the input sequence for protein A was based on the free structure sequence in the PPI-Hotspot+PDB$^{BM(1.1)}$, whereas that for protein B was retrieved in its entirety from the UniProtKB (*UniProt Consortium, 2018*) as the binding region in protein B was unknown. Based on the AMBER-relaxed model structure with the highest pTM score, interface residues were defined as residues of protein A with ≥1 atom within a 5 Å cutoff of any protein B atom.

## Experimental verification of predicted eEF2 PPI-hot spots: Plasmid construction

The predicted PPI-hot spots and PPI-*non*hot spots were mutated using the QuikChange Site-Directed Mutagenesis Kit (Stratagene). The pcDNA3.1-flag-meEF2 plasmid was used as the PCR template, and the sets of sense and antisense primers for mutagenesis are listed in *Figure 2—source data 3*. All constructs were sequenced to confirm the mutations. The shRNA clones, #1 TRCN0000047908 (GCGATCATGAATTTCAAGAAA) and #2: TRCN0000047910 (GCAGTACCTCAACGAGATCAA), against human eEF2 mRNA were obtained from the RNAi Core Facility (Academia Sinica).

### Cell lines

HEK-293T cells (# CRL-3216) and HeLa cells (# CCL-2) were obtained from American Type Culture Collection (ATCC).

## Testing eEF2-CPEB2 interactions using co-IP and reciprocal co-IP

HEK-293T cells (ATCC, # CRL-3216) were cultured in DMEM with 10% fetal bovine serum (FBS). For reciprocal co-IP, the 8 µg DNA mixture containing 3 µg myc-CPEB2 and 5 µg flag-eEF2 (or a negative control, GFP) plasmids, generated in the previous study (*Chen and Huang, 2012*), was transfected into a 10 cm dish of 293T cells using Lipofectamine 2000. We transfected more flag-eEF2 plasmid DNA than myc-CPEB2 plasmid because myc-CPEB2 is expressed more abundantly than flag-eEF2. Overnight transfected cells were lysed in 500 µl IP buffer (20 mM HEPES, pH 7.4, 100 mM NaCl, 1 mM MgCl₂, 0.1% TritonX-100, 10% glycerol, 0.5 mM DTT, 1× protease inhibitor cocktail, and 100 µg/ml RNaseA) and centrifuged at 10,000 × *g* for 3 min at 4°C. The supernatant was equally divided and incubated with Protein G beads bound with myc (Abcam, 9E10 clone) or flag (Sigma-Aldrich, F1804) antibody for 3 hr at 4°C to respectively pull down myc-CPEB2 and flag-eEF2. The beads were washed five times with 300 µl IP buffer. If myc-CPEB2 and flag-eEF2 interact, myc-CPEB2 can co-precipitate with flag-eEF2 on flag antibody beads, whereas flag-eEF2 can co-precipitate with myc-CPEB2 on myc antibody beads. GFP was used as a negative control to ensure that the signals on the beads were

caused by binding between flag-eEF2 and myc-CPEB2. The precipitated proteins were separated on a sodium dodecyl sulfate polyacrylamide gel electrophoresis (SDS-PAGE) for western blot analysis. Similarly, for the initial co-IP screening, the 4 µg DNA mixture containing 1.5 µg myc-CPEB2 and 2.5 µg flag-eEF2 (or a negative control, EGFP) plasmids was transfected into a 6 cm dish of 293T cells, harvested in 200 µl IP buffer, and immunoprecipitated using flag antibody-bound beads.

## Functional impact of eEF2 mutants on HIF-1α and global protein synthesis

HeLa cells (ATCC, # CCL-2) were cultured in DMEM with 10% FBS. Each 6 cm plate of HeLa cells was transfected with 2 µg human eEF2 knockdown plasmid and 2 µg flag-meEF2 wild-type/mutant plasmid. Overnight transfected cells were selected with 0.5 µg/ml puromycin and 600 µg/ml G418 for 3 days to knock down endogenous eEF2 and maintain the expression of flag-meEF2, respectively. The selected cells were incubated with 10 µM MG132 for 4 hr before harvesting for western blotting of HIF-1α or replaced with 2 ml Met/Cys-lacking DMEM with 1% FBS and 60 µCi $^{35}$S-Met/Cys (PerkinElmer, cat# NEG772002MC) for 2 hr before separation on an SDS-PAGE. Antibodies used are CPEB2, generated in house; HIF-1α (NB100-134) from Novus; eEF2 (SC-13004) from Santa Cruz Biotechnology; and β-actin (A5441) form Sigma-Aldrich.

## Acknowledgements

This research was supported by Academia Sinica (AS-IA-107-L03) and the Ministry of Science and Technology (MOST-98-2113M-001-011), Taiwan.

## Additional information

### Competing interests

Jon D Wright, Carmay Lim: Affiliated with Immunwork, Inc; the author has no other competing interests to declare. The other authors declare that no competing interests exist.

### Funding

| Funder | Grant reference number | Author |
| --- | --- | --- |
| Academia Sinica | AS-IA-107-L03 | Carmay Lim |
| Ministry of Education | MOST-98-2113-M-001-011 | Carmay Lim |
| Institute of Biomedical Sciences, Academia Sinica | | Yi-Shuian Huang |

The funders had no role in study design, data collection and interpretation, or the decision to submit the work for publication.

### Author contributions

Yao Chi Chen, Conceptualization, Data curation, Software, Formal analysis, Validation, Investigation, Visualization, Methodology, Writing – original draft; Karen Sargsyan, Software, Validation, Investigation, Methodology, Writing – original draft; Jon D Wright, Software, Investigation; Yu-Hsien Chen, Data curation, Formal analysis, Investigation; Yi-Shuian Huang, Formal analysis, Funding acquisition, Validation, Investigation, Writing – original draft; Carmay Lim, Conceptualization, Resources, Supervision, Investigation, Methodology, Project administration, Writing – review and editing

### Author ORCIDs

Yao Chi Chen ⓘ http://orcid.org/0000-0003-0205-6078
Yu-Hsien Chen ⓘ https://orcid.org/0000-0002-0475-1042
Yi-Shuian Huang ⓘ https://orcid.org/0000-0001-9000-7748
Carmay Lim ⓘ https://orcid.org/0000-0001-9077-7769

Reviewer #1 (Public review): https://doi.org/10.7554/eLife.96643.3.sa1

Author response https://doi.org/10.7554/eLife.96643.3.sa2

## Additional files

### Supplementary files

• MDAR checklist

• Source data 1. Dataset of experimentally confirmed PPI-hot spots and PPI-nonhot spots with free protein structures.

• Source data 2. Dataset of experimentally confirmed PPI-hot spots with both free and bound protein structures.

### Data availability

All data generated or analyzed during this study are included in the manuscript and supporting files. The PPI-HotspotID program is available at https://github.com/wrigjz/ppihotspotid/ (**Wright, 2024**). A web server to perform PPI-hotspot predictions and the dataset comprising 414 experimentally known PPI-hot spots and 504 PPI-nonhot spots are available at https://ppihotspotid.limlab.dnsalias.org/.

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
