## [Editor Report · eLife assessment]

The article presents a machine-learning method to predict protein hotspot residues. The validation is **incomplete**, along with the misinterpretation of the results with other current methods like FTMap.

---

## [Referee Report · Reviewer #1 (Public review)]

The paper describes a program developed to identify PPI-hot spots using the free protein structure and compares it to FTMap and SPOTONE, two webservers that they consider as competitive approaches to the problem. We appreciate the effort in providing a new webserver that can be tested by the community but we continue to have major concerns:

(1) The comparison to the FTMap program is problematic. The authors misinterpret the article they refer to, i.e., Zerbe et al. "Relationship between hot spot residues and ligand binding hot spots in protein-protein interfaces" J. Chem. Inf. Model. 52, 2236-2244, (2012). FTMap identifies hot spots that bind small molecular ligands. The Zerbe et al. article shows that such hot spots tend to interact with hot spot residues on the partner protein in a protein-protein complex (emphasis on "partner"). Thus, the hot spots identified by FTMap are not the hot spots defined by the authors. In fact, because the Zerbe paper considers the partner protein in a complex, the results cannot be compared to the results of Chen et al. This difference is missed by the authors, and hence the comparison of the FTMap is invalid.

(2) Chen et al. use a number of usual features in a variety of simple machine-learning methods to identify hot spot residues. This approach has been used in the literature for more than a decade. Although the authors say that they were able to find only FTMap and SPOTONE as servers, there are dozens of papers that describe such a methodology. Some examples are given here: (Higa and Tozzi, 2009; Keskin, et al., 2005; Lise, et al., 2011; Tuncbag, et al., 2009; Xia, et al., 2010). There are certainly more papers. Thus, while the web server is a potentially useful contribution, the paper does not provide a fundamentally novel approach.

---

## [Author Response]

The following is the authors’ response to the original reviews.

**eLife assessment**
The manuscript presents a machine-learning method to predict protein hotspot residues. The validation is incomplete, along with the misinterpretation of the results with other current methods like FTMap.

We believe that validation is complete: The two most common techniques for testing and validating machine-learning methods are to split the dataset into either (1) a training set and a test set with a fixed ratio (e.g., 70% for training and 30% for testing) or (2) multiple subsets/folds; i.e., cross-validation. We did not employ a training set to train the model and a separate test set to evaluate its performance, as Reviewer 2 assumed. Instead, we employed cross-validation, as it helps reduce the variability in performance estimates compared to a single training/test split, and utilizes the entire dataset for training and testing, making efficient use of the limited data. Each fold was used once as a test set and the remaining folds as the training set - this process was repeated for each fold and the model's performance was measured using the F1 score. We had listed the mean validation F1 score in Table 1.

We have clarified our comparison with FTMAP - see reply to point 1 of reviewer 1 below.

**Public Reviews:**

**Reviewer #1 (Public Review):**
Summary:The paper describes a program developed to identify PPI-hot spots using the free protein structure and compares it to FTMap and SPOTONE, two webservers that they consider as competitive approaches to the problem. On the positive side, I appreciate the effort in providing a new webserver that can be tested by the community but have two major concerns as follows.(1) The comparison to the FTMap program is wrong. The authors misinterpret the article they refer to, i.e., Zerbe et al. "Relationship between hot spot residues and ligand binding hot spots in protein-protein interfaces" J. Chem. Inf. Model. 52, 2236-2244, (2012). FTMap identifies hot spots that bind small molecular ligands. The Zerbe et al. article shows that such hot spots tend to interact with hot spot residues on the partner protein in a protein-protein complex (emphasis on "partner"). Thus, the hot spots identified by FTMap are not the hot spots defined by the authors. In fact, because the Zerbe paper considers the partner protein in a complex, the results cannot be compared to the results of Chen et al. This difference is missed by the authors, and hence the comparison of the FTMap is invalid. I did not investigate the comparison to SPOTONE, and hence have no opinion.

Brenke et al. (Bioinformatics 2009 25: 621-627), who developed FTMAP, defined hot spots as regions of the binding surface that “contribute a disproportionate amount to the binding free energy”. Kozakov et al. (Proc. Natl. Acad. Sci. 2011:108, 13528-1353) used unbound protein structures as input to FTMap to predict binding hot spots for protein-protein interactions (PPIs), which are defined as regions (so-called consensus sites) on a protein surface that bind multiple probe clusters − the main hot spot is the largest consensus site binding the largest number of probe clusters.

Zerbe et al. (J. Chem. Inf. Model. 2012:52, 2236) noted that a consensus “site is expected to be important in any interaction that involves that region of the target independent of any partner protein.” They showed that for hot spot residues found by Ala scanning not only overlapped with the probe ligands but also form consensus sites, as shown in Figure 4. They stated that “A residue can also be identified as a hot spot by alanine scanning if it contributes to creating such a favorable binding environment by being among the residues forming a consensus site on the protein to which it belongs.”

To clarify the comparison with FTmap in the revised version, we have added the following sentence in the Abstract on p. 3:

“We explored the possibility of detecting PPI-hot spots using (i) FTMap in the PPI mode, which identifies hot spots on protein-protein interfaces from the free protein structure, and (ii) the interface residues predicted by AlphaFold-Multimer.”

We have added the following sentences in the Introduction section on p. 4:

“We explored the possibility of detecting PPI-hot spots using the FTMap server in the PPI mode, which identifies hot spots on protein-protein interfaces from free protein structures.45 These hot spots are identified by consensus sites − regions that bind multiple probe clusters.42,45,59 Such regions are deemed to be important for any interaction involving that region of the target, independent of partner protein.42 PPIhot spots were identified as residues in van der Waals (vdW) contact with probe ligands within the largest consensus site containing the most probe clusters.”

and in the Results section on p. 5:

“Given the free protein structure, PPI-HotspotID and SPOTONE53 predict PPI-hot spots based on a probability threshold (> 0.5). FTMap, in the PPI mode, detects PPIhot spots as consensus sites/regions on the protein surface that bind multiple probe clusters.59 Residues in vdW contact with probe molecules within the largest consensus site were compared with PPI-hotspotID/SPOTONE predictions.”

(2) Chen et al. use a number of usual features in a variety of simple machine-learning methods to identify hot spot residues. This approach has been used in the literature for more than a decade. Although the authors say that they were able to find only FTMap and SPOTONE as servers, there are dozens of papers that describe such a methodology. Some examples are given here: (Higa and Tozzi, 2009; Keskin, et al., 2005; Lise, et al., 2011; Tuncbag, et al., 2009; Xia, et al., 2010). There are certainly more papers. Thus, while I consider the web server as a potentially useful contribution, the paper does not provide a fundamentally novel approach.

Our paper introduces several novel elements in our approach:

(1) Most PPI-hot spot prediction methods employ PPI-hotspots where mutations decrease protein binding free energy by > 2 kcal/mol (J. Chem. Inf. Model. 2022, 62, 1052). In contrast, our method incorporates not only PPI-hot spots with such binding free energy changes, but also those whose mutations have been curated in UniProtKB to significantly impair/disrupt PPIs. Because our method employs the largest collection of experimentally determined PPI-hot spots, it could uncover elusive PPI-hot spots not within binding interfaces, as well as potential PPI-hot spots for other protein partners (see point 3 below).

(2) Whereas most machine-learning methods for PPI-hot spot prediction focus on features derived from (i) primary sequences or (ii) protein-protein complexes, we introduce novel features such as per-residue free energy contributions derived from unbound protein structures. We further revealed the importance of one of our novel features, namely, the gas-phase energy of the target protein relative to its unfolded state and provided the physical basis for its importance. For example, PPI-hot spots can enhance favorable enthalpic contributions to the binding free energy through hydrogen bonds or van der Waals contacts across the protein’s interface. This makes them energetically unstable in the absence of the protein’s binding partner and solvent; hence providing a rationale for the importance of the gas-phase energy of the target protein relative to its unfolded state.

(3) As a result of these novel elements, our approach, PPI-HotspotID, could identify many true positives that were not detected by FTMap or SPOTONE (see Results and Figure 1). Previous methods generally predict residues that make multiple contacts across the proteinprotein interface as PPI-hot spots. In contrast, PPI-HotspotID can detect not only PPI-hot spots that make multiple contacts across the protein-protein interface, but also those lacking direct contact with the partner protein (see Discussion).

(4) Unlike most machine-learning methods which require feature customization, data preprocessing, and model optimization, our use of AutoGluon’s AutoTabular module automates data preprocessing, model selection, hyperparameter optimization, and model evaluation. This automation reduces the need for manual intervention.

We have revised and added the following sentences on p. 9 in the Discussion section to highlight the novelty of our approach:

“Here, we have introduced two novel elements that have helped to identify PPI-hot spots using the unbound structure. First, we have constructed a dataset comprising 414 experimentally known PPI-hot spots and 504 nonhot spots, and carefully checked that PPI-hot spots have no mutations resulting in ΔΔG^bind^ < 0.5 kcal/mol, whereas nonhot spots have no mutations resulting in ΔΔG^bind^ ≥ 0.5 kcal/mol or impact binding in immunoprecipitation or GST pull-down assays (see Methods). In contrast, SPOTONE53 employed nonhot spots defined as residues that upon alanine mutation resulted in ΔΔG^bind^ < 2.0 kcal/mol. Notably, previous PPI-hot spot prediction methods did not employ PPIhot spots whose mutations have been curated to significantly impair/disrupt PPIs in UniProtKB (see Introduction). Second, we have introduced novel features derived from unbound protein structures such as the gas-phase energy of the target protein relative to its unfolded state.”

Strengths:A new web server was developed for detecting protein-protein interaction hot spots.Weaknesses:The comparison to FTMap results is wrong. The method is not novel.

See reply to points 1 and 2 above.

**Reviewer #2 (Public Review):**
Summary:The paper presents PPI-hotspot a method to predict PPI-hotspots. Overall, it could be useful but serious concerns about the validation and benchmarking of the methodology make it difficult to predict its reliability.Strengths:Develops an extended benchmark of hot-spots.Weaknesses:(1) Novelty seems to be just in the extended training set. Features and approaches have been used before.

The novelty of our approach extends beyond just the expanded training set, as summarized in our reply to Reviewer #1, point 2 above. To our knowledge, previous studies did not leverage the gas-phase energy of the target protein relative to its unfolded state for detecting PPI-hot spots from unbound structures. Previous studies did not automate the training and validation process. In contrast, we used AutoGluon’s AutoTabular module to automate the training of (i) individual “base” models, including LightGBM, CatBoost, XGBoost, random forests, extremely randomized trees, neural networks, and K-nearest neighbours, then (ii) multiple “stacker” models. The predictions of multiple “stacker” models were fed as inputs to additional higher layer stacker models in an iterative process called multi-layer stacking. The output layer used ensemble selection to aggregate the predictions of the stacker models. To improve stacking performance, AutoGluon used all the data for both training and validation through repeated k-fold bagging of all models at all layers of the stack, where k is determined by best precision. This comprehensive approach, including repeated k-fold bagging of all models at all layers of the stack, sets our methodology apart from previous studies, including SPOTONE (see Methods).

(2) As far as I can tell the training and testing sets are the same. If I am correct, it is a fatal flaw.

The two most common techniques for testing and validating machine-learning methods are to split the dataset into either (1) a training set and a test set with a fixed ratio (e.g., 70% for training and 30% for testing) or (2) multiple subsets/folds; i.e., cross-validation. We did not employ a training set to train the model and a separate test set to evaluate its performance. Instead, we employed cross-validation, where the model was trained and evaluated multiple times. Each fold was used once as a test set and the remaining folds serve as the training set - this process was repeated for each fold. For each test set, we assessed the model's performance using the F1 score. We had listed the mean validation F1 score in Table 1 in the original manuscript. Cross-validation helps reduce the variability in performance estimates compared to a single training/test split. It also utilizes the entire dataset for training and testing, making efficient use of the limited data. We have clarified this on p. 14 in the revised version:

“AutoGluon was chosen for model training and validation due to its robustness and userfriendly interface, allowing for the simultaneous and automated exploration of various machine-learning approaches and their combinations. Instead of using a single training set to train the model and a separate test set to evaluate its performance, we employed cross-validation, as it utilizes the entire dataset for both training and testing, making efficient use of the limited data on PPI-hot spots and PPI-nonhot spots. AutoGluonTabular automatically chose a random partitioning of our dataset into multiple subsets/folds for training and validation. Notably, the training and validation data share insignificant homology, as the average pairwise sequence identity in our dataset is 26%. Each fold was used once as a test set, while the remaining folds served as the training set. For each test set, the model's performance was measured using the F1 score.”

(3) Comparisons should state that: SPOTONE is a sequence (only) based ML method that uses similar features but is trained on a smaller dataset. FTmap I think predicts binding sites, I don't understand how it can be compared with hot spots. Suggesting superiority by comparing with these methods is an overreach.

In the Introduction on page 3, we had already stated that:

“SPOTONE53 predicts PPI-hot spots from the protein sequence using residue-specific features such as atom type, amino acid (aa) properties, secondary structure propensity, and mass-associated values to train an ensemble of extremely randomized trees. The PPIhot spot prediction methods have mostly been trained, validated, and tested on data from the Alanine Scanning Energetics database (ASEdb)55 and/or the Structural Kinetic and Energetic database of Mutant Protein Interactions (SKEMPI) 2.0 database.56”

On p. 4, we have clarified how we used FTMAP to detect hot spots - see reply to Reviewer #1, point 1.

“We explored the possibility of detecting PPI-hot spots using the FTMap server in the PPI mode, which identifies hot spots on protein-protein interfaces from free protein structures.45 These hot spots are identified by consensus sites − regions that bind multiple probe clusters.42,45,59 Such regions are deemed to be important for any interaction involving that region of the target, independent of partner protein.42 PPI-hot spots were identified as residues in van der Waals (vdW) contact with probe ligands within the largest consensus site containing the most probe clusters.”

(4) Training in the same dataset as SPOTONE, and then comparing results in targets without structure could be valuable.

We think that the dataset used by SPOTONE is not as “clean” as ours since SPOTONE employed nonhot spots defined as aa residues that upon alanine mutation resulted in ΔΔG^bind^ < 2.0 kcal/mol. In contrast, we define nonhot spots as residues whose mutations resulted in protein ΔΔG^bind^ changes < 0.5 kcal/mol. Moreover, we carefully checked that the nonhot spots have no mutations resulting in ΔΔG^bind^ changes ≥ 0.5 kcal/mol or impact binding in immunoprecipitation or GST pull-down assays (see Methods). We cannot compare results in targets without structure because we require the free protein structure to compute the perresidue free energy contributions.

(5) The paper presents as validation of the prediction and experimental validation of hotspots in human eEF2. Several predictions were made but only one was confirmed, what was the overall success rate of this exercise?

We did not test all predicted PPI-hot spots but only the PPI-hot spot with the highest probability of 0.67 (F794) and 7 other predicted PPI-hot spots that were > 12 Å from F794 as well as 4 predicted PPI-nonhot spots. Among the 13 predictions tested, F794 and the 4 predicted nonhot spots were confirmed to be correct.

**Recommendations for the authors:**

**Reviewer #1 (Recommendations For The Authors):**
Remove the comparison to FTMap, and find a more appropriate reference method, even if it requires installing programs rather than using the available web servers.

We have clarified comparison to FTMap in the revised ms - see our reply above.